REGISTERED REPORT PROTOCOL

# Prospective observational study of nutritional/immunologic indices as predictive biomarkers for the response to anti-PD-1 drugs in non-small cell lung cancer (ICI-PREDICT study)

**Shinkichi Takamori**[1], **Taro Ohba**[2], **Mototsugu Shimokawa**[3], **Taichi Matsubara**[1], **Naoki Haratake**[1], **Naoko Miura**[1], **Ryo Toyozawa**[1], **Masafumi Yamaguchi**[1], **Takashi Seto**[1], **Mitsuhiro Takenoyama**[1]*

1 Department of Thoracic Oncology, National Hospital Organization Kyushu Cancer Center, Notame, Minami-ku, Fukuoka, Japan, 2 Department of Surgery and Science, Graduate School of Medical Sciences, Kyushu University, Maidashi, Higashi-ku, Fukuoka, Japan, 3 Department of Biostatistics, Yamaguchi University Graduate School of Medicine, Minamiogushi, Ube-shi, Yamaguchi, Japan

* takenoyama.m@gmail.com

## Abstract

Immune checkpoint inhibitors (ICIs) targeting programmed cell death-1 (PD-1) and programmed cell death-ligand 1 (PD-L1) have markedly improved the prognosis of many patients with advanced non-small cell lung cancer (NSCLC). However, the relationship between the patient's nutritional/immunologic status and the outcomes of ICI treatment remains unclear. In previous retrospective studies, we reported that the controlling nutritional status (CONUT) score, skeletal muscle area, and neutrophil-to-lymphocyte ratio were independent predictors of the response of NSCLC patients to anti-PD-1 drugs. The aim of this prospective multi-center study is to investigate the clinical impact of pre-treatment nutritional/immunologic indices and early post-treatment changes in the indices on treatment outcomes in advanced NSCLC. The main inclusion criteria are: (1) stage IV NSCLC, or stage III NSCLC not applicable for definitive chemoradiotherapy; (2) treatment with ICIs (monotherapy or combined with chemotherapy) as first-line therapy; and (3) available data on PD-L1 expression on tumor cells. A total of 300 patients will be enrolled prospectively. Enrollment will begin in 2020 and the final analyses will be completed by 2025.

## Introduction

Lung cancer, of which non-small cell lung cancer (NSCLC) accounts for 85–90% of cases, is the deadliest cancer worldwide [1]. The recent development of immune checkpoint inhibitors (ICIs) targeting programmed cell death-1 (PD-1) and programmed cell death-ligand 1 (PD-L1) has contributed to improvements in the prognosis of many patients with advanced NSCLC [2, 3]. In a trial of the anti-PD-1 antibody pembrolizumab [3], subgroup analysis

**Data Availability Statement:** All relevant data from this study will be made available upon study completion.

**Funding:** The publication fee was funded by Grantsin-Aid for Scientific Research (JSPS KAKENHI) Grant Number 20839542.

**Competing interests:** The authors declare no conflicts of interest in association with the present study.

indicated that PD-L1 expression on tumor cells is a potential predictive factor of the response to anti-PD-1 antibodies [4]. However, the relationship between responses to ICIs and key host factors, such as nutritional and immunologic statuses, has not been fully investigated.

Several measures have been used as nutritional indices in retrospective studies, including the controlling nutritional status (CONUT) score, which is based on peripheral lymphocyte counts and serum albumin and cholesterol concentrations, and skeletal muscle area measured by computed tomography (CT) or other imaging modalities [5–7]. In our previous single center retrospective study of NSCLC patients receiving pembrolizumab, patients with poor nutritional status, defined as a CONUT score of ≥3, had significantly shorter progression-free survival and overall survival compared with patients with good nutritional status (for disease progression or death: hazard ratio [HR] 0.33, 95% confidence interval [CI] 0.10–0.97; for death: HR 0.15, 95% CI 0.02–0.73) [8]. In that analysis, the CONUT score was an independent predictive factor for response to pembrolizumab. A preliminary study of NSCLC patients receiving ICIs showed that patients with low skeletal muscle areas had a lower overall response rate (9.1% vs. 40.0%) and 1-year progression-free survival rate (10.1% vs. 38.1%) compared with patients with normal skeletal muscle areas [9]. In addition to the nutritional indices, a recent meta-analysis has shown that the pre-treatment neutrophil-to-lymphocyte ratio is a candidate predictive factor for the efficacy of ICIs in cancer patients [10]. In our study of pembrolizumab, we found that neutrophil-to-lymphocyte ratio was an independent predictive factor for overall survival (HR 0.25, 95% CI 0.05–0.98) in patients with advanced NSCLC [8]. Thus, the clinical implications of pre-treatment nutritional/immunological statuses in NSCLC patients receiving ICIs has attracted great attention. However, these findings remain to be validated in prospective and/or multi-center studies with larger sample sizes. The aim of this prospective study is to investigate the clinical impact of pre-treatment nutritional/immunologic indices on the outcomes of patients with advanced NSCLC who are treated with ICIs. The relationship between early post-treatment changes in nutritional/immunologic indices and outcomes will also be analyzed in the study.

## Methods

### Study design

This is a prospective multi-center observational study of patients with advanced NSCLC. The nutritional/immunologic indices, which include CONUT score, skeletal muscle area, and neutrophil-to-lymphocyte ratio will be measured before treatment initiation, twice after treatment initiation (first and second response evaluations) to determine the clinical implications of changes observed in the early post-treatment phase, and a fourth time at the end of first-line treatment. The study design is shown in Fig 1. The protocol has been approved by National Hospital Organization Central Review Board.

### Study population

All patients who satisfy the inclusion and exclusion criteria (Table 1) will be included. In brief, NSCLC patients (i) with stage IV disease or with stage III disease not applicable for definitive chemoradiotherapy, (ii) who receive ICIs (monotherapy or combined with chemotherapy) as first-line therapy, and (iii) available data on PD-L1 expression on tumor cells will be included. Patients who harbor any oncogene mutations, have other synchronous cancers, have a history of cancer immunotherapy, or who are considered to be unsuitable for enrollment in the investigator's judgment, will be excluded. We are planning to recruit participant hospitals in all parts of Japan from National Hospital Organization. Each hospital will recruit eligible participants. A sample size and power calculation before study initiation was not performed because

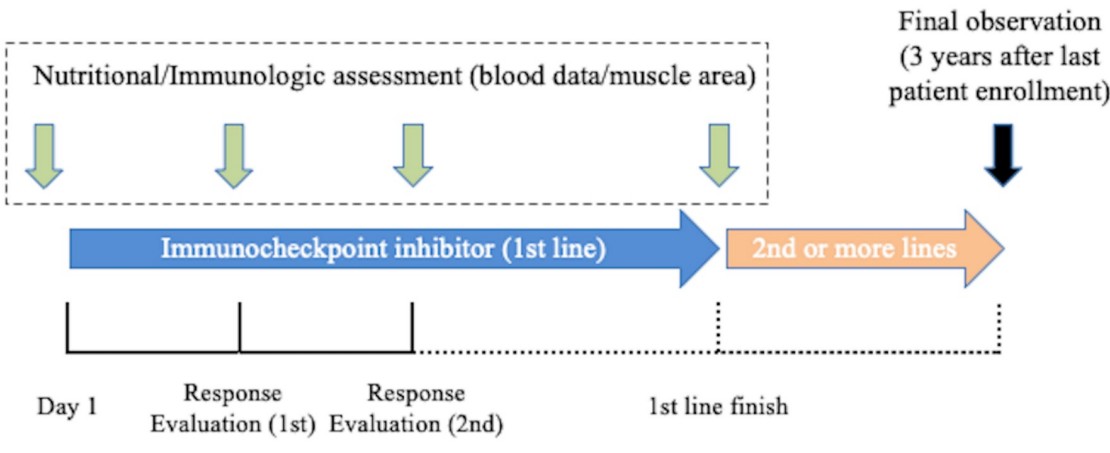

**Fig 1. Study design.**

this is an exploratory observational study for deciding the appropriate cut-off value of the nutritional/immunologic indices. A total of 300 patients will be enrolled prospectively over 2 years.

## PD-L1 expression

PD-L1 expression on tumor cells will be analyzed by immunohistochemistry using one of several anti-PD-L1 antibodies, including clones 22C3, 28–8, SP142, and SP263 [11]. Of note, one of the inclusion criteria for this study is "available data on PD-L1 expression on tumor cells (tumor proportion score)," which will allow us to compare the clinical significance of the patient's nutritional/immunologic status with that of tumor PD-L1 expression.

**Table 1. Key patient eligibility criteria.**

| |
|---|
| Inclusion criteria |
| 1) Pathologically confirmed non-small cell lung cancer (cytology or histology) |
| 2) Stage IV disease or Stage III disease not applicable for definitive chemoradiotherapy |
| 3) Treatment with checkpoint inhibitors (monotherapy or combined with chemotherapy) as 1st line therapy |
| 4) Available programmed cell death-ligand 1 expression on tumor cells (tumor proportion score) |
| 5) Any target lesions designated by Response Evaluation Criteria in Solid Tumors version 1.1 |
| 6) Minimum age of 20 years old |
| 7) Written informed consent from the patient |
| Exclusion criteria |
| 1) Any oncogene mutation (*epidermal growth receptor factor, anaplastic lymphoma kinase, c-ros oncogene 1, RET, BRAF*) |
| 2) Difficulty in evaluating pre-treatment nutritional/immunologic |
| 3) Other synchronous cancer |
| 4) Active infection with HIV, hepatitis B, or hepatitis C virus |
| 5) History of cancer immunotherapy |
| 6) Considered to be unsuitable for enrolment by the investigator's judgment |

**Table 2. Assessment of patient nutritional status by the CONUT score.**

| Factors | Range and Score | | | |
|---|---|---|---|---|
| Albumin (g/dL) | ≥3.50 | 3.00–3.49 | 2.50–2.99 | <2.50 |
| Score | 0 | 2 | 4 | 6 |
| Cholesterol (mg/dL) | ≥180 | 140–179 | 100–139 | <100 |
| Score | 0 | 1 | 2 | 3 |
| Lymphocyte count (/mm$^3$) | ≥1600 | 1200–1599 | 800–1199 | <800 |
| Score | 0 | 1 | 2 | 3 |

CONUT: Controlling nutritional status = albumin score + cholesterol score + lymphocyte score.

## Evaluation of nutritional/immunologic indices

The CONUT score is defined as shown in Table 2 [6]. We are planning to categorize the CONUT score into high CONUT vs. low CONUT groups. The time-dependent receiver operating characteristic (ROC) curve for the prediction of progression-free survival will be used when we set the cut-off value of CONUT score. In addition, we are going to categorize the changes in nutritional/immunologic indices into "deteriorated group" and "improved group." When we set the cut-off value, the time-dependent ROC curve for the prediction of progression-free survival will be used. The skeletal muscle area will be measured by CT with OsiriX® software (version 5.8; Geneva, Switzerland) using Hounsfield Units with a threshold from −29 to +150 [12]. The following muscles at the L3 level will be included in the skeletal muscle area assessment: psoas, erector spinae, quadratus lumborum, transversus abdominis, external and internal obliques, and rectus abdominis [12]. The skeletal muscle area will be normalized by dividing the cross-sectional areas (cm$^2$) at the L3 level by the height squared (m$^2$). The neutrophil-to-lymphocyte ratio will be calculated as the ratio of neutrophil to lymphocyte counts, as measured by standard procedures.

## Statistical analysis

The relationship between the patient's nutritional/immunologic status and the clinical factors will be analyzed using a Student's $t$ test, Pearson's $\chi^2$ test, and Fisher's exact test where appropriate. Responses to anti-PD-1 agents will be determined according to the Response Evaluation Criteria in Solid Tumors (version 1.1). The overall survival will be defined as the time (in months) from the first day of immunotherapy until death from any cause. The progression-free survival will be defined as the time (in months) from the first day of treatment until progression. The Kaplan–Meier method will be used to estimate survival probabilities, and differences will be analyzed using the log-rank test. Univariate and multivariable analyses will be conducted with the log-rank test and a proportional hazards model. In multivariable logistic analysis, the backward elimination method will be used. In brief, the model will be run with all the variables, and one variable with the highest $P$ value will be excluded. The model will be run again with the other variables, and one variable with the highest $P$ value will be excluded repeatedly to retain only those with $P < 0.05$. Cox proportional hazard regression models will be used to calculate hazard ratios for positive risk factors with a similar backward elimination method. The "timeROC" R package (https://cran.r-project.org/web/packages/timeROC/index.html) will be used to compare the area under the curve of ROC curves for the prediction of progression-free and overall survivals for each nutritional/immunologic status. We are planning to adjust for age, sex, smoking status, disease severity (advanced disease vs. recurrent disease), and performance status in the analyses. $P$ values <0.05 will be considered statistically

significant. All statistical analyses will be performed using JMP software version 14.0 (SAS Institute Inc., Cary, NC, USA).

## Ethical considerations

Written informed consent will be obtained from all patients before any inclusion procedures. The study will be conducted according to the principles of the Declaration of Helsinki. The protocol has been approved by National Hospital Organization Central Review Board.

## Discussion and conclusion

Little is known about the relationship between pre-treatment and early post-treatment changes in the nutritional/immunologic status and clinical outcomes in patients with advanced NSCLC receiving ICIs. We hope that this prospective multi-center study will shed light on the clinical implications, including prognosis, of the nutritional/immunologic status for this patient population.

## Acknowledgments

We thank Anne M. O'Rourke, PhD, from Edanz Group (www.edanzediting.com/ac) for editing a draft of this manuscript.

## Author Contributions

**Conceptualization:** Shinkichi Takamori, Taro Ohba.

**Data curation:** Shinkichi Takamori, Taichi Matsubara, Naoki Haratake, Ryo Toyozawa.

**Formal analysis:** Shinkichi Takamori, Mototsugu Shimokawa.

**Funding acquisition:** Shinkichi Takamori.

**Investigation:** Takashi Seto.

**Methodology:** Shinkichi Takamori, Naoko Miura.

**Supervision:** Masafumi Yamaguchi.

**Writing – original draft:** Shinkichi Takamori.

**Writing – review & editing:** Mitsuhiro Takenoyama.

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
