## [Decision Letter · Decision Letter 0]

10 Nov 2020

PONE-D-20-05915

Prospective Observational Study of Nutritional/Immunologic Indices as Predictive Biomarkers for the Response to Anti-PD-1 Drugs in Non-small Cell Lung Cancer (ICI-PREDICT study)

PLOS ONE

Dear Dr. Tokenoyama:

Thank you for submitting your manuscript to PLOS ONE. After careful consideration, we feel that it has merit but does not fully meet PLOS ONE’s publication criteria as it currently stands. Therefore, we invite you to submit a revised version of the manuscript that addresses the points raised during the review process.

Both reviewers suggested some major changes that can be made to improve the work.  Please address these in a revision.

We look forward to receiving your revised manuscript.

Kind regards,

Gayle E. Woloschak, PhD

Academic Editor

PLOS ONE

Additional Editor Comments:

One reviewer rejected this, the other suggested major revision. Please attempt to address all of the concerns raised as much as possible and document it in a cover letter.

Journal Requirements:

2. Thank you for including your ethics statement: 'The protocol has been approved by the institutional review boards of the participating hospitals.'

(a) Please amend your current ethics statement to include the full name of the ethics committee/institutional review board(s) that approved your specific study.  

(b) Once you have amended this/these statement(s) in the Methods section of the manuscript, please add the same text to the “Ethics Statement” field of the submission form (via “Edit Submission”).

3. In your Methods section, please provide additional information about the planned participant recruitment method. Please ensure you have provided sufficient details to replicate the analyses such as:  a) a description of how participants will be recruited, and b) descriptions of where participants will be recruited and where the research will take place.

4. Please provide a sample size and power calculation in the Methods, or discuss the reasons for not performing one before study initiation.

5. To comply with PLOS ONE submission guidelines, in your Methods section, please provide additional information regarding your statistical analyses, including the set level of statistical significance that will be used in the study. For more information on PLOS ONE's expectations for statistical reporting, please see https://journals.plos.org/plosone/s/submission-guidelines.#loc-statistical-reporting.

7. PLOS requires an ORCID iD for the corresponding author in Editorial Manager on papers submitted after December 6th, 2016. Please ensure that you have an ORCID iD and that it is validated in Editorial Manager. To do this, go to ‘Update my Information’ (in the upper left-hand corner of the main menu), and click on the Fetch/Validate link next to the ORCID field. This will take you to the ORCID site and allow you to create a new iD or authenticate a pre-existing iD in Editorial Manager. Please see the following video for instructions on linking an ORCID iD to your Editorial Manager account: https://www.youtube.com/watch?v=_xcclfuvtxQ

Reviewers' comments:

Reviewer's Responses to Questions

**Comments to the Author**

1. Does the manuscript provide a valid rationale for the proposed study, with clearly identified and justified research questions?

Reviewer #1: Yes

Reviewer #2: Yes

2. Is the protocol technically sound and planned in a manner that will lead to a meaningful outcome and allow testing the stated hypotheses?

Reviewer #1: Partly

Reviewer #2: Yes

3. Is the methodology feasible and described in sufficient detail to allow the work to be replicable?

Reviewer #1: Yes

Reviewer #2: Yes

4. Have the authors described where all data underlying the findings will be made available when the study is complete?

Reviewer #1: Yes

Reviewer #2: No

5. Is the manuscript presented in an intelligible fashion and written in standard English?

Reviewer #1: Yes

Reviewer #2: Yes

6. Review Comments to the Author

You may also provide optional suggestions and comments to authors that they might find helpful in planning their study.

Reviewer #1: Abstract:

Please replace "multi-institutional" with "multi-centre" (or "multi-center"); it's a more readily recognised term for what I believe you are referring to.

Introduction:

For the same reason as above please replace "single-institutional" with "single centre" (or "single center").

I appreciate the necessity for abbreviations in some cases, but I think that the manuscript would be easier to follow if abbreviations were restricted to where they are absolutely necessary. Terms such as overall survival, progression-free survival, skeletal muscle area and neutrophil-to-lymphocyte ratio need not be abbreviated in the text (but may be abbreviated in tables and figures).

Methods:

The information in lines 82 to 86 seems to have been repeated in lines 89 to 94, it is unclear why.

Some justification of the sample size of 300 should be provided, if possible a formal sample size calculation.

More detail is needed in the description of the statistical analysis. The main explanatory variable is the CONUT score; it is unclear how the K-M and log-rank methods will be applied with this continuous (or at best, ordinal) measure - will it be categorised? If so, what thresholds will be used for the categorisation.

It is also not clear how the repeated measures of nutritional/immunologic indices (measured before administration of ICI, first and second response evaluation days, last day of therapy) will be used in the analysis; this needs to be clarified. It should also be clarified whether a multivariable (one outcome variable) or multivariate (several outcome variables in one model) regression will be conducted, and the terminology appropriately used.

The analysis should also clarify which, if any, factors which influence outcomes, such as disease severity (cancer stage), age of patient, type of treatment whether mono or combination therapy) will be adjusted for in the analysis.

Reviewer #2: This manuscript describes the trial plan of a prospective studies of nutrients and immunological indices as a predictor for response to anti-PD-1 therapy among non-small cell lung cancer patients. The manuscript was nicely written with a clear background and research design. At the same time, because it is a protocol, it has limited if any new knowledge, which may not be best for the readers for PLOS one as the reported protocol may be interested to specialized readers, such as clinicians. So the content may not be a good fit for the journal. One minor writing issue: table 1 was not included.

7. PLOS authors have the option to publish the peer review history of their article (what does this mean?). If published, this will include your full peer review and any attached files.

Reviewer #1: No

Reviewer #2: No

---

## [Author Response · Author response to Decision Letter 0]

20 Jan 2021

January 20th, 2021

PLoS ONE Editorial Office

Dear Editors,

 We are grateful for your valuable comments about our manuscript entitled "Prospective Observational Study of Nutritional/Immunologic Indices as Predictive Biomarkers for the Response to Anti-PD-1 Drugs in Non-small Cell Lung Cancer (ICI-PREDICT study)." (PONE-D-20-05915). We have done our best to improve the manuscript based on the reviewers’ comments. In addition, we changed a sentence to the Financial Disclosure part as follows: The publication fee was funded by Grants-in-Aid for Scientific Research (JSPS KAKENHI) Grant Number 20839542. We are hereby sending the responses to the comments from the reviewers, along with the revised version of the manuscript.

Respectfully yours,

Mitsuhiro Takenoyama

Department of Thoracic Oncology, National Hospital Organization Kyushu Cancer Center, 3-1-1 Notame, Minami-ku, Fukuoka, 811-1395, Japan

Tel.: +81 92 541 3231; Fax: +81 92 551 4585

E-mail: takenoyama.m@gmail.com

Responses to the Comments

Comments:

1. Thank you for your response. Before we proceed, could you please confirm whether any pilot data are reported within this manuscript?

If so, please indicate where the data can be accessed and provide all necessary URLs, DOIs, and/or contact information.

* If not, please update your Data Availability Statement to read: "All relevant data from this study will be made available upon study completion."

Responses to the Comments:

1. I confirm that no pilot data are reported within our manuscript. So, I updated my Data Availability Statement as follows: "All relevant data from this study will be made available upon study completion."

 

Reviewers' comments:

Reviewer #1: 

Abstract:

1) Please replace "multi-institutional" with "multi-centre" (or "multi-center"); it's a more readily recognised term for what I believe you are referring to.

Introduction:

2) For the same reason as above please replace "single-institutional" with "single centre" (or "single center").

3) I appreciate the necessity for abbreviations in some cases, but I think that the manuscript would be easier to follow if abbreviations were restricted to where they are absolutely necessary. Terms such as overall survival, progression-free survival, skeletal muscle area and neutrophil-to-lymphocyte ratio need not be abbreviated in the text (but may be abbreviated in tables and figures).

Methods:

4) The information in lines 82 to 86 seems to have been repeated in lines 89 to 94, it is unclear why.

5) Some justification of the sample size of 300 should be provided, if possible a formal sample size calculation.

6) More detail is needed in the description of the statistical analysis. The main explanatory variable is the CONUT score; it is unclear how the K-M and log-rank methods will be applied with this continuous (or at best, ordinal) measure - will it be categorised? If so, what thresholds will be used for the categorisation.

7) It is also not clear how the repeated measures of nutritional/immunologic indices (measured before administration of ICI, first and second response evaluation days, last day of therapy) will be used in the analysis; this needs to be clarified. It should also be clarified whether a multivariable (one outcome variable) or multivariate (several outcome variables in one model) regression will be conducted, and the terminology appropriately used.

8) The analysis should also clarify which, if any, factors which influence outcomes, such as disease severity (cancer stage), age of patient, type of treatment whether mono or combination therapy) will be adjusted for in the analysis.

Responses to Reviewer #1:

1) Thank you very much for your suggestion. We replaced "multi-institutional" with "multi-center." (page 3, line 34; page 5, line 76; page 5, line 84; page 11, line 184)

2) We replaced "single-institutional" with "single center." (page 4, line 58)

3) Thank you very much for your comments. We agree that abbreviations should be restricted to where they are absolutely necessary. Overall survival, progression-free survival, skeletal muscle area, and neutrophil-to-lymphocyte ratio were changed not to be abbreviated in the text.

4) Thank you very much for your pointing it out. We deleted the figure explanations since they were redundant (page 6, lines 94-99).

5) Thank you very much for your comment. Regarding a formal sample size calculation, we did not perform it because this is an exploratory observational study for deciding the appropriate cut-off value of the nutritional/immunologic indices. We have already consulted a biostatistician (Mototsugu Shimokawa; one of the co-authors) about the need for calculation, and confirmed that there is no need to perform it. However, we added the discussion about the reasons for not performing the calculation to the manuscript (page 7, line 111-113).

6) Thank you very much for your pointing out. We agree that detailed information is needed in the description of the statistical analysis. We are planning to categorize the CONUT score into high CONUT vs. low CONUT groups. The time-dependent ROC curve for the prediction of progression-free survival will be used when we set the cut-off value of CONUT score. We added these comments to our manuscript (page 8, line 128-131).

7) Thank you very much for your comments. We agree that it is not clear how the repeated measures of nutritional/immunologic indices will be used in the analysis. We are planning to categorize the changes in nutritional/immunologic indices into “deteriorated group” and “improved group.” When we set the cut-off value, the time-dependent ROC curve for the prediction of progression-free survival will be used. In addition, a multivariable regression analyses will be conducted (page 8, line 132-134). Accordingly, the “multivariate” was modified to “multivariable.” (page 10, line 158)

8) We are planning to adjust for age, sex, smoking status, disease severity (advanced disease vs. recurrent disease), and performance status in the analyses. We added the information to our manuscript (page 10, line 168-170).

Reviewer #2: This manuscript describes the trial plan of a prospective studies of nutrients and immunological indices as a predictor for response to anti-PD-1 therapy among non-small cell lung cancer patients. The manuscript was nicely written with a clear background and research design. At the same time, because it is a protocol, it has limited if any new knowledge, which may not be best for the readers for PLOS one as the reported protocol may be interested to specialized readers, such as clinicians. So the content may not be a good fit for the journal. One minor writing issue: table 1 was not included.

Response to Reviewer #2:

Thank you for your comments. It seems that we have included Table 1 in the manuscript.

---

## [Decision Letter · Decision Letter 1]

4 Oct 2021

Prospective Observational Study of Nutritional/Immunologic Indices as Predictive Biomarkers for the Response to Anti-PD-1 Drugs in Non-small Cell Lung Cancer (ICI-PREDICT study)

PONE-D-20-05915R1

Dear Dr. Takenoyama:

We’re pleased to inform you that your manuscript has been judged scientifically suitable for publication and will be formally accepted for publication once it meets all outstanding technical requirements.

Kind regards,

Gayle E. Woloschak, PhD

Section Editor

PLOS ONE

Additional Editor Comments (optional):

Thank you for addressing the concerns raised.

Reviewers' comments:

Reviewer's Responses to Questions

**Comments to the Author**

1. Does the manuscript provide a valid rationale for the proposed study, with clearly identified and justified research questions?

Reviewer #1: Yes

Reviewer #3: Yes

2. Is the protocol technically sound and planned in a manner that will lead to a meaningful outcome and allow testing the stated hypotheses?

Reviewer #1: Yes

Reviewer #3: Yes

3. Is the methodology feasible and described in sufficient detail to allow the work to be replicable?

Reviewer #1: Yes

Reviewer #3: Yes

4. Have the authors described where all data underlying the findings will be made available when the study is complete?

Reviewer #1: Yes

Reviewer #3: Yes

5. Is the manuscript presented in an intelligible fashion and written in standard English?

Reviewer #1: Yes

Reviewer #3: Yes

6. Review Comments to the Author

You may also provide optional suggestions and comments to authors that they might find helpful in planning their study.

Reviewer #1: In line 116-118 and 120-121, if possible please give a bit more detail about how the ROC curve will be used to set the cut-off value, e.g. what criteria will be used to make this decision.

Reviewer #3: The manuscript titled ''Prospective Observational Study of Nutritional/Immunologic Indices as Predictive Biomarkers for the Response to Anti-PD-1 Drugs in Non-small Cell Lung Cancer (ICIPREDICT study)'' was very well written and addressed all the previous reviewers comments.

7. PLOS authors have the option to publish the peer review history of their article (what does this mean?). If published, this will include your full peer review and any attached files.

Reviewer #1: No

Reviewer #3: **Yes: **Prasanta Kumar Nayak

---

## [Editor Report · Acceptance letter]

14 Oct 2021

PONE-D-20-05915R1 

Prospective Observational Study of Nutritional/Immunologic Indices as Predictive Biomarkers for the Response to Anti-PD-1 Drugs in Non-small Cell Lung Cancer (ICI-PREDICT study) 

Dear Dr. Takenoyama:

I'm pleased to inform you that your manuscript has been deemed suitable for publication in PLOS ONE. Congratulations! Your manuscript is now with our production department. 

Kind regards, 

on behalf of

Dr. Gayle E. Woloschak 

Section Editor

PLOS ONE